# Computationally-efficient initialisation of GPs: The generalised variogram method

**Felipe Tobar**                     *ftobar@uchile.cl*
*Initiative for Data & AI*
*Universidad de Chile*

**Elsa Cazelles**                    *elsa.cazelles@irit.fr*
*CNRS, IRIT*
*Université de Toulouse*

**Taco de Wolff**                 *tacodewolff@gmail.com*
*Inria Chile*

**Reviewed on OpenReview:** *https://openreview.net/forum?id=slsAQHpS7n*

## Abstract

We present a computationally-efficient strategy to initialise the hyperparameters of a Gaussian process (GP) avoiding the computation of the likelihood function. Our strategy can be used as a pretraining stage to find initial conditions for maximum-likelihood (ML) training, or as a standalone method to compute hyperparameters values to be plugged in directly into the GP model. Motivated by the fact that training a GP via ML is equivalent (on average) to minimising the KL-divergence between the true and learnt model, we set to explore different metrics/divergences among GPs that are computationally inexpensive and provide hyperparameter values that are close to those found via ML. In practice, we identify the GP hyperparameters by projecting the empirical covariance or (Fourier) power spectrum onto a parametric family, thus proposing and studying various measures of discrepancy operating on the temporal and frequency domains. Our contribution extends the variogram method developed by the geostatistics literature and, accordingly, it is referred to as the generalised variogram method (GVM). In addition to the theoretical presentation of GVM, we provide experimental validation in terms of accuracy, consistency with ML and computational complexity for different kernels using synthetic and real-world data.

## 1 Introduction

Gaussian processes (GPs) are Bayesian nonparametric models for time series praised by their interpretability and generality. Their implementation, however, is governed by two main challenges. First, the choice of the covariance kernel, which is usually derived from first principles or expert knowledge and thus may result in complex structures that hinder hyperparameter learning. Second, the cubic computational cost of standard, maximum-likelihood-based, training which renders the exact GP unfeasible for more than a few thousands observations. The GP community actively targets these issues though the development of robust, computationally-efficient training strategies. Though these advances have facilitated the widespread use of GP models in realistic settings, their success heavily depends on the initialisation of the hyperparameters.

In practice, initialisation either follows from expert knowledge or time-consuming stochastic search. This is in sharp contrast with the main selling point of GPs, that is, being agnostic to the problem and able to freely learn from data. To provide researchers and practitioners with an automated, general-application and cost-efficient initialisation methodology, we propose to learn the hyperparameters by approximating the empirical covariance by a parametric function using divergences between covariances that are inexpensive

to compute. This approach is inspired by the common practice, such as when one computes some statistics (e.g., mean, variance, discrete Fourier transform) to determine initial values for the kernel hyperparameters. In the geostatistics literature, a method that follows this concept is the Variogram (Cressie, 1993; Chiles & Delfiner, 1999), which is restricted to particular cases of kernels and divergences. Therefore, we refer to the proposed methodology as Generalised Variogram Method (GMV) in the sense that it extends the application of the classic methodology to a broader scenario that includes general stationary kernels and metrics, in particular, Fourier-based metrics. Though our proposal is conceptually applicable to an arbitrary input/output dimension, we focus on the scalar-input scalar-output case and leave the extension to general dimensions as future work.[1]

The contributions of our work are of theoretical and applied nature, and are summarised as follows:

- a novel, computationally-efficient, training objective for GPs as an alternative to maximum likelihood, based on a projection of sample covariance (or power spectrum) onto a parametric space;
- a particular instance of the above objective, based on the Wasserstein distance applied to the power spectrum, that is convex and also admits a closed-form solution which can thus be found in a single step;
- a study of the computational cost of the proposed method and its relationship to maximum likelihood, both in conceptual and empirical terms;
- experimental validation of the proposed method assessing its stability with respect to initial conditions, confirming its linear computational cost, its ability to train GPs with a large number of hyperparameters and its advantages as initialiser for maximum-likelihood training.

## 2 Preliminaries

### 2.1 Motivation

Let us consider $y \sim \mathcal{GP}(0, K)$ and its realisation $\boldsymbol{y} = [y_1, \ldots, y_n] \in \mathbb{R}^n$ at times $\boldsymbol{t} = [t_1, \ldots, t_n] \in \mathbb{R}^n$. The kernel $K$ is usually learnt by choosing a family $\{K_\theta\}_{\theta \in \Theta}$ and optimising the log-likelihood

$$l(\theta) = -\frac{1}{2} \operatorname{Tr}\left(\mathbf{K}_\theta^{-1} \boldsymbol{y} \boldsymbol{y}^\top\right) - \frac{1}{2} \log |\mathbf{K}_\theta| - \frac{n}{2} \log 2\pi, \tag{1}$$

with respect to $\theta \in \Theta$, where we used the cyclic property of the trace, and defined $\mathbf{K}_\theta \stackrel{\text{def}}{=} K_\theta(\boldsymbol{t})$ according to $[\mathbf{K}_\theta]_{ij} = K_\theta(t_i - t_j), i, j \in \{1, \ldots, n\}$. Defining $\mathbf{K} \stackrel{\text{def}}{=} K(\boldsymbol{t}) = \mathbb{E} \boldsymbol{y} \boldsymbol{y}^\top$, we note that

$$\mathbb{E} l(\theta) = -\frac{1}{2} \operatorname{Tr}\left(\mathbf{K}_\theta^{-1} \mathbf{K}\right) - \frac{1}{2} \log |\mathbf{K}_\theta| - \frac{n}{2} \log 2\pi. \tag{2}$$

Notice that, up to terms independent of $\theta$, equation 2 is equivalent to the negative Kullback-Leibler divergence (NKL) between the (zero-mean) multivariate normal distributions $\mathcal{N}(0, \mathbf{K})$ and $\mathcal{N}(0, \mathbf{K}_\theta)$ given by

$$D_{\text{NKL}}(\mathcal{N}(0, \mathbf{K}) \| \mathcal{N}(0, \mathbf{K}_\theta)) = -\frac{1}{2}\left(\operatorname{Tr}\left(\mathbf{K}_\theta^{-1} \mathbf{K}\right) + \log \frac{|\mathbf{K}_\theta|}{|\mathbf{K}|} - n\right), \tag{3}$$

which, with a slight abuse of notation, can be expressed as a function of only the covariances as $D_{\text{NKL}}(\mathbf{K}, \mathbf{K}_\theta) \stackrel{\text{def}}{=} D_{\text{NKL}}(\mathcal{N}(0, \mathbf{K}) \| \mathcal{N}(0, \mathbf{K}_\theta))$.

This reveals that learning a GP by maximising $l(\theta)$ in equation 1 can be understood (in expectation) as minimising the KL between the $\boldsymbol{t}$-marginalisations of the true process $\mathcal{GP}(0, K)$ and a candidate process $\mathcal{GP}(0, K_\theta)$. This motivates the following remark.

**Remark 1.** *Since maximum-likelihood learning of GPs has a cubic computational cost but it is (on average) equivalent to minimising a KL divergence, what other divergences or distances, computationally cheaper than the likelihood, can be considered for learning GPs?*

---

[1]For illustration and completeness, we incorporate a toy experiment featuring 5-dimensional input data in Sec. 5.7.

## 2.2 Divergences over covariance functions

We consider zero-mean stationary GPs.[2] The stationary requirement allows us to i) aggregate observations in time when computing the covariance, and ii) compare covariances in terms of their (Fourier) spectral content. Both perspectives will be present throughout the text, thus, we consider two types of divergences over covariances: i) **temporal** ones, which operate directly to the covariances, and ii) **spectral** ones, which operate over the the Fourier transform of the covariances, i.e., the GP's power spectral density (PSD). As our work relies extensively on concepts of spectral analysis and signal processing, a brief introduction to the subject is included in Appendix A.

Though we can use most metrics (e.g., $L_1$, $L_2$) on both domains, the advantage of the spectral perspective is that it admits density-specific divergences as it is customary in signal processing (Basseville, 1989). Bregman divergences, which include the Euclidean, KL and Itakura-Saito (IS) (Itakura, 1968), are *vertical* measures, i.e, they integrate the point-wise discrepancy between densities across their support. We also consider *horizontal* spectral measures, based on the minimal-cost to *transport* the mass from one distribution—across the support space—onto another. This concept, known as optimal transport (OT) (Villani, 2009) has only recently been considered for comparing PSDs using, e.g., the 1- and 2-Wasserstein distances, denoted $W_1$ and $W_2$ (Cazelles et al., 2021; Henderson & Solomon, 2019). See Appendix B for definitions of vertical and horizontal divergences.

## 2.3 Related work

A classic notion of kernel dissimilarity in the machine learning community is that of *kernel alignment*, a concept introduced by Cristianini et al. (2001) which measures the agreement between two kernels or a kernel and a task; this method has been mostly applied for kernel selection in SVM-based regression and classification. This notion of dissimilarity is based on the Frobenius inner product between the Gram matrices of each kernel given a dataset—see (Cristianini et al., 2001, Def. 1). Though, in spirit, this concept is related to ours in that the kernel is learnt by minimising a discrepancy measure, we take a signal-processing inspired perspective and exploit the fact that, for stationary GPs, kernels are single-input covariance functions and thus admit computationally-efficient discrepancy measures. In addition to the reference above, the interested reader is referred to (Cortes et al., 2012) for the *centred* kernel alignment method.

Within the GP community, two methodologies for accelerated training can be identified. The first one focuses directly on the optimisation procedure by, e.g., avoiding inverses (van der Wilk et al., 2020), or derivatives (Rios & Tobar, 2018), or even by parallelisation; combining these techniques has allowed to process even a million datapoints (Wang et al., 2019). A second perspective is that of sparse GP approximations using *pseudo-inputs* (Quinonero-Candela & Rasmussen, 2005), with particular emphasis on variational methods (Titsias, 2009). The rates of convergence of sparse GPs has been studied by Burt et al. (2019) and the hyperparameters in this approach have also been dealt with in a Bayesian manner by Lalchand et al. (2022). Sparse GPs have allowed for fitting GPs to large datasets (Hensman et al., 2013), training non-parametric kernels (Tobar et al., 2015), and implementing deep GPs (Damianou & Lawrence, 2013; Salimbeni & Deisenroth, 2017). Perhaps the work that is closest to ours in the GP literature is that of Liu et al. (2020), which trains a neural network to learn the mapping from datasets to GP hyperparameters thus avoiding the computation of the likelihood during training. However, the authors state that "*training on very large datasets consumes too much GPU memory or becomes computationally expensive due to the kernel matrix size and the cost of inverting the matrix*". This is because they still need to compute the kernel matrix during training of the net, while we bypass that calculation altogether.

The Wasserstein distance has been used to compare laws of GPs (Masarotto et al., 2019; Mallasto & Feragen, 2017), and applied to kernel design, in particular to define GPs (Bachoc et al., 2018) and deep GPs (Popescu et al., 2022) over the space of probability distributions. Cazelles et al. (2021) proposed a distance between time series based on the Wasserstein, termed the Wasserstein-Forier distance, and showed its application

---

[2]We do so for convenience in computation since non-zero-mean GPs can be understood as a mixture of a zero-mean GP and a parametric regression model, where the GP models the residuals of the parametric part. See Section 2.7 in Rasmussen & Williams (2005) for a discussion.

to GPs. However, despite the connection between GPs and the Wasserstein distance established by these works, they have not been implemented to train (or initialise) GPs.

In geostatistics, the variogram function (Cressie, 1993; Chiles & Delfiner, 1999) is defined as the variance of the difference of a process $y$ at two locations $t_1$ and $t_2$, that is, $\gamma(t_1 - t_2) = \mathbb{V}[y(t_1) - y(t_2)]$. The variogram is computed by choosing a parametric form for $\gamma(t)$ and then fit it to a cloud of points (sample variogram) using least squares. Common variogram functions in the literature include exponential and Gaussian ones, thus drawing a natural connection with GP models. Furthermore, when the process $y$ is stationary and isotropic (or one-dimensional) as in the GP models considered here, the variogram and the covariance $K(t)$ follow the relationship $\gamma(t) = K(0) - K(t)$, therefore, given a kernel function the corresponding variogram function can be clearly identified (and vice versa). The way in which the variogram is fitted in the geostatistics literature is what inspires the methodology proposed here: we match parametric forms of the covariance (or the PSD) to their corresponding sample approximations in order to find appropriate values for the kernel hyperparameters. Also, as we explore different distances for the covariance and PSD beyond the Euclidean one (least squares) we refer to our proposal as the *Generalised Variogram Method* (GVM).

GVM complements the literature in a way that is orthogonal to the above developments. We find the hyperparameters of a GP in a likelihood-free manner by minimising a loss function operating directly on the sample covariance or its Fourier transform. As we will see, GVM is robust to empirical approximations of the covariance or PSDs, admits arbitrary distances and has a remarkably low computational complexity.

## 3   A likelihood-free covariance-matching strategy for training GPs

**Overview of the section.** As introduced in Section 2.1, our objective is to approximate the ground-truth kernel $K$ with a parametric kernel $K_{\theta^*}$; to this end we will rely on an empirical data-driven estimate of the kernel denoted $\hat{K}_n$. We will proceed by matching $\hat{K}_n$ with a parametric form $K_{\theta_n^*}$ using metrics in the temporal domain, i.e., solving equation 7, or in the spectral domain matching the Fourier transform of $\hat{K}_n$, denoted $\hat{S}_n = \mathcal{F}\hat{K}_n$, with a parametric form $S_{\theta_n^*}$, i.e., solving equation 8. In the following, we present the Fourier pairs $K_\theta$ and $S_\theta$, the estimators $\hat{K}_n$ and $\hat{S}_n$, and the metrics considered for the matching. We then present an explicit case for location-scale families, and we finally give theoretical arguments for the proposed learning method.

### 3.1   Fourier pairs $K_\theta$ and $S_\theta$ and their respective estimators

Let us consider the zero-mean stationary process $y \sim \mathcal{GP}(0, K_\theta)$ with covariance $K_\theta$ and hyperparameter $\theta \in \Theta$. If the covariance $K_\theta$ is integrable, Bochner's Theorem (Bochner, 1959) states that $K_\theta$ and the process' power spectral density (PSD), denoted $S_\theta$, are *Fourier pairs*, that is,

$$S_\theta(\xi) = \mathcal{F}\{K_\theta\} \stackrel{\text{def}}{=} \int_{\mathbb{R}} K_\theta(t)e^{-j2\pi\xi t}dt, \tag{4}$$

where $j$ is the imaginary unit and $\mathcal{F}\{\cdot\}$ denotes the Fourier transform.

Since zero-mean stationary GPs are uniquely determined by their PSDs, any distance defined on the space of PSDs can be "lifted" to the space covariances and then to that of GPs. Therefore, we aim at learning the hyperparameter $\theta \in \Theta$ by building statistics in either the temporal space (sample covariances) or in spectral space (sample PSDs).

First, we consider the following statistic $\hat{K}_n$ in order to approximate the ground-truth kernel $K$.

**Definition 1.** *Let $y \in \mathbb{R}$ be a zero mean stochastic process over $\mathbb{R}$ with observations $\boldsymbol{y} = [y_1, \ldots, y_n] \in \mathbb{R}^n$ at times $\boldsymbol{t} = [t_1, \ldots, t_n] \in \mathbb{R}^n$. The empirical covariance of $y$ is given by*

$$\hat{K}_n(t) = \sum_{i,j=1}^{n} \frac{y_i y_j \mathbf{1}_{t=t_i-t_j}}{Card\{t|t = t_i - t_j\}}. \tag{5}$$

The estimator $\hat{S}_n(t)$ of the PSD $S$ then simply corresponds to applying the Fourier transform to the empirical kernel $\hat{K}_n$ in equation 5, that is

$$\hat{S}_n(\xi) = \int_{\mathbb{R}} \hat{K}_n(t)e^{-j2\pi\xi t}dt. \tag{6}$$

Another usual choice for estimating the PSD is the Periodogram $\hat{S}_{\mathrm{Per}}$ (Schuster, 1900). Though $\hat{S}_{\mathrm{Per}}$ is asymptotically unbiased ($\forall \xi, \mathbb{E}\hat{S}_{\mathrm{Per}}(\xi) \to S(\xi)$), it is inconsistent, i.e., its variance does not vanish when $n \to \infty$ (Stoica & Moses, 2005)[Sec. 2.4.2]. Luckily, the variance of $\hat{S}_{\mathrm{Per}}(\xi)$ can be reduced via *windowing* and the Welch/Bartlett techniques which produce (asymptotically) consistent and unbiased estimates of $S(\xi), \forall \xi$ (Stoica & Moses, 2005).

### 3.2 Fourier-based covariance divergences

The proposed method builds on two types of (covariance) similarity. First, those operating directly on $\hat{K}_n$ and $K_\theta$, known as **temporal** divergences, which include the $L_1$ and $L_2$ distances. Second, those operating over the Fourier transforms of $\hat{K}_n$ and $K_\theta$, that is $\hat{S}_n$ and $S_\theta$, known as **spectral** divergences. In the temporal case, we aim to find the hyperparameters of $y$ by projecting $\hat{K}_n(t)$ in equation 5 onto the parametrised family $\mathcal{K} = \{K_\theta, \theta \in \Theta\}$. That is, by finding $\theta^*$ such that $K_\theta(\cdot)$ is *as close as possible* to $\hat{K}_n(t)$, i.e.,

$$\theta_n^* = \underset{\theta \in \Theta}{\arg\min}\, D(\hat{K}_n, K_\theta), \tag{7}$$

where the function $D(\cdot, \cdot)$ is the chosen criterion for similarity. Akin to the strategy of learning the hyperparameters of the GP by matching the covariance, we can learn the hyperparameters by projecting an estimator of the PSD, namely $\hat{S}_n$ in equation 6, onto a parametric family $\mathcal{S} = \{S_\theta, \theta \in \Theta\}$, that is,

$$\theta_n^* = \underset{\theta \in \Theta}{\arg\min}\, D_F(\hat{S}_n, S_\theta), \tag{8}$$

where $D_F(\cdot, \cdot)$ is a divergence operating on the space of PSDs.

**Remark 2.** *Since the map $K_\theta \to S_\theta$ is one-to-one, equation 7 and equation 8 are equivalent when $D_F(\cdot, \cdot) = D(\mathcal{F}\{\cdot\}, \mathcal{F}\{\cdot\})$ and $\mathcal{S} = \mathcal{F}\{\mathcal{K}\}$.*

We will consider parametric families $\mathcal{S}$ with explicit inverse Fourier transform, since this way $\theta$ parametrises both the kernel and the PSD and can be learnt in either domain. These families include the Dirac delta, Cosine, Square Exponential (SE), Student's $t$, Sinc, Rectangle, and their mixtures; see Figure 10 in Appendix A for an illustration of some of these PSDs and their associated kernels.

**Remark 3** (Recovering kernel parameters from PSD parameters)**.** *For a parametric Fourier pair (K, S), the Fourier transform induces a map between the kernel parameter space and the PSD parameter space. For kernel/PSD families considered in this work this map will be bijective, which allows us to compute the estimated kernel parameters from the estimated PSD parameters (and vice versa); see Table 1 for two examples of parametric kernels and PSDs. Furthermore, based on this bijection we refer as $\theta$ to both kernel and PSD parameters.*

### 3.3 A particular case with explicit solution: location-scale family of PSDs and $2$-Wasserstein distance

Of particular relevance to our work is the 2-Wasserstein distance ($W_2$) and *location-scale* families of PSDs, for which the solution of equation 8 is completely determined through first order conditions.

**Definition 2** (Location–scale)**.** *A family of one-dimensional integrable PSDs is said to be of location-scale type if it is given by*

$$\left\{ S_{\mu,\sigma}(\xi) = \frac{1}{\sigma} S_{0,1}\left(\frac{\xi - \mu}{\sigma}\right), \mu \in \mathbb{R}, \sigma \in \mathbb{R}_+ \right\}, \tag{9}$$

*where $\mu \in \mathbb{R}$ is the location parameter, $\sigma \in \mathbb{R}_+$ is the scale parameter and $S_{0,1}$ is the prototype of the family.*

For arbitrary prototypes $S_{0,1}$, location-scale families of PSDs are commonly found in the GP literature. For instance, the SE, Dirac delta, Student's $t$, Rectangular and Sinc PSDs, correspond to the Exp-cos, Cosine,

Table 1: Location-scale PSDs (left) and their covariance kernel (right) used in this work. We have denoted by $t$ and $\xi$ the time and frequency variables respectively.

| **PSD** | $S_{\mu,\sigma}(\xi)$ | Prototype $S_{0,1}(\xi)$ | **Kernel** | $K_{\mu,\sigma}(t)$ | $K_{0,1}(t)$ |
|---|---|---|---|---|---|
| Square-exp | $\exp\left(-\left(\frac{\xi-\mu}{\sigma}\right)^2\right)$ | $\exp(-\xi^2)$ | Exp-cos | $\sqrt{\pi}\sigma\exp(-\sigma^2\pi^2t^2)\cos(2\pi\mu t)$ | $\sqrt{\pi}\exp(-\pi^2t^2)$ |
| Rectangular | $\text{rect}\left(\frac{\xi-\mu}{\sigma}\right)$ | $\text{rect}(\xi)$ | Sinc | $\sigma\text{sinc}(\sigma t)\cos(2\pi\mu t)$ | $\text{sinc}(t)$ |

Laplace, Sinc, and Rectangular kernels respectively. Location-scale families do not, however, include kernel mixtures, which are also relevant in our setting and will be dealt with separately. Though the prototype $S_{0,1}$ might also be parametrised (e.g., with a *shape* parameter), we consider those parameters to be fixed and only focus on $\theta := (\mu, \sigma)$ for the rest of this section. Table 1 shows the two families of location-scale PSDs (and their respective kernels) that will be used throughout our work.

**Remark 4.** *Let us consider a location-scale family of distributions with prototype $S_{0,1}$ and an arbitrary member $S_{\mu,\sigma}$. Their quantile (i.e., inverse cumulative) functions, denoted $Q_{0,1}$ and $Q_{\mu,\sigma}$ respectively, obey*

$$Q_{\mu,\sigma}(p) = \mu + \sigma Q_{0,1}(p). \tag{10}$$

The linear expression in equation 10 is pertinent in our setting and motivates the choice of the 2-Wasserstein distance $W_2$ to compare members of the location-scale family of PSDs. This is because for arbitrary one-dimensional distributions $S_1$ and $S_2$, $W_2^2(S_1, S_2)$ can be expressed according to:

$$W_2^2(S_1, S_2) = \int_0^1 (Q_1(p) - Q_2(p))^2 dp, \tag{11}$$

where $Q_1$ and $Q_2$ denote the quantiles of $S_1$ and $S_2$ respectively. We are now in position to state the first main contribution of our work.

**Theorem 1.** *If $\mathcal{S}$ is a location-scale family with prototype $S_{0,1}$, and $S$ is an arbitrary PSD, the minimiser $(\mu^*, \sigma^*)$ of $W_2(S, S_{\mu,\sigma})$ is unique and given by*

$$\mu^* = \int_0^1 Q(p)dp \qquad and \qquad \sigma^* = \frac{1}{\int_0^1 Q_{0,1}^2(p)dp} \int_0^1 Q(p)Q_{0,1}(p)dp, \tag{12}$$

*where $Q$ is the quantile function of $S$.*

*Proof Sketch.* The proof follows from the fact that $W_2^2(S, S_{\mu,\sigma})$ is convex both on $\mu$ and $\sigma$, which is shown by noting that its Hessian is positive via Jensen's inequality. Then, the first order conditions give the solutions in equation 12. The complete proof can be found in Appendix C. □

**Remark 5.** *Although integrals of quantile functions are not usually available in closed-form, computing equation 12 is straightforward. First, $Q_{0,1}(p)$ is known for a large class of normalised prototypes, including square exponential and rectangular functions. Second, $\mu^* = \mathbb{E}_{x\sim S}[x]$ and $\int_0^1 Q_{0,1}^2(p)dp = \mathbb{E}_{x\sim S}[x^2]$, where x is a random variable with probability density $S$. Third, both integrals are one-dimensional and supported on $[0,1]$, thus numerical integration is inexpensive and precise, specially when $S$ has compact support.*

As pointed out by Cazelles et al. (2021), $W_2^2(S, S_\theta)$ is in general non-convex in $\theta$, however, for the particular case of the location-scale family of PSDs and parameters $(\mu, \sigma)$, convexity holds. Since this family includes usual kernel choices in the GP literature, Theorem 1 guarantees closed form solution for the optimisation problem in equation 8 and thus can be instrumental for learning GPs without computing (the expensive) likelihood function.

### 3.4 Learning from data

Learning the hyperparameters through the optimisation objective in equation 7 or equation 8 is possible provided that the statistics $\hat{K}_n$ and $\hat{S}_n$ converge to $K$ and $S$ respectively. We next provide theoretical

results on the convergence of the optimal minimiser $\theta_n^*$. The first result, Proposition 1, focuses on the particular case of the 2-Wasserstein distance and location-scale families, presented in Section 3.3.

**Proposition 1.** *Let $S_\theta$ be a location-scale family, $S$ the ground truth PSD and $D_F = W_2^2$. Then $\mathbb{E} W_2^2(S, \hat{S}_n) \to 0$ implies that the empirical minimiser $\theta_n^*$ in equation 8 converges to the true minimiser $\theta^* = \arg\min_{\theta \in \Theta} W_2^2(S, S_\theta)$, meaning that $\mathbb{E}|\theta_n^* - \theta^*| \to 0$.*

*Proof Sketch.* First, in the location-scale family we have $\theta = (\mu, \sigma)$. Then, the solutions in Theorem 1 allow us to compute upper bounds for $|\mu^* - \mu_n^*|$ and $|\sigma^* - \sigma_n^*|$ via Jensen's and Hölder's inequalities, both of which converge to zero as $\mathbb{E} W_2^2(S, \hat{S}_n) \to 0$. The complete proof can be found in Appendix C. $\square$

The second result, Proposition 2, deals with the more general setting of arbitrary parametric families, the distances $L_1, L_2, W_1, W_2$ and either temporal and frequency based estimators. To cover both cases, we will denote $\hat{f}_n$ to refer to either $\hat{K}_n$ or $\hat{S}_n$. Let us also consider the parametric function $f_\theta \in \{f_\theta | \theta \in \Theta\}$ and the ground-truth function $f$ which denotes either the ground-truth covariance or ground-truth PSD.

**Proposition 2.** *For general parametric families $\{f_\theta | \theta \in \Theta\}$, the empirical solution $\theta_n^*$ converges a.s. to the true solution $\theta^*$ under the following sufficient and stronger conditions: (i) $D = W_r$ or $L_r, r = 1, 2$, (ii) $D(\hat{f}_n, f) \xrightarrow[n \to \infty]{a.s.} 0$; (iii) $\theta_n \xrightarrow[n \to \infty]{} \theta \iff D(f_{\theta_n}, f_\theta) \to 0$; and (iv) the parameter space $\Theta$ is compact.*

*Proof.* This result follows as a particular case from Theorem 2.1 in Bernton et al. (2019), where the authors study general $r$-Wasserstein distance estimators for parametric families of distributions for empirical measures, under the notion of $\Gamma$-convergence or, equivalently, epi-convergence. The proof for the $L_r, r = 1, 2$ case is similar to that of $W_r$. $\square$

## 4 Practical considerations

This section is dedicated to the implementation of GVM. We first discuss the calculation of the estimators $\hat{K}_n$ and $\hat{S}_n$ and their complexity, we then present the special case of location-scale family and spectral 2-Wasserstein loss. The last subsection is dedicated to the noise variance parameter and the relationship between GVM and ML solutions.

### 4.1 Computational complexity of the estimators $\hat{K}_n$ and $\hat{S}_n$

First, for temporal divergences (operating on the covariance) we can modify the statistic in equation 5 using *binning*, that is, by averaging values for which the lags are similar; this process is automatic in the case of evenly-sampled data and widely used in discrete-time signal processing. This allows to reducing the amount of summands in $\hat{K}_n$ from $\frac{n(n+1)}{2}$ to an order $n$ or even lower if the range of the data grows beyond the length of the correlations of interests.

Second, the Periodogram $\hat{S}_{\text{Per}}$ can be computed at a cost of $\mathcal{O}(nk)$ using bins, where $n$ is the number of observations and $k$ the number of frequency bins; in the evenly-sampled case, one could set $k = n$ and apply the fast Fourier transform at a cost $\mathcal{O}(n \log n)$. However, for applications where the number of datapoints greatly exceeds the required frequency resolution, $k$ can be considered to be constant, which results in a cost linear in the observations $\mathcal{O}(n)$.

### 4.2 The location-scale family and 2-Wasserstein distance

Algorithm 1 presents the implementation of GVM in the case of location-scale family of PSDs $\{S_\theta\}_{\theta \in \Theta}$ and 2-Wasserstein loss presented in Section 3.3.

---
**Algorithm 1** GVM - Spectral loss $W_2$ & ($\{S_\theta\}_{\theta \in \Theta}$ **is** location-scale)
---
**Require:** $t_i, y_i, i = 1, \ldots, n$
**Require:** location-scale PSD family $\{S_\theta\}_{\theta \in \Theta}$ (e.g., Gaussians)
   Define a grid over frequency space $\mathbf{g} = \{\xi_0, \xi_1, \ldots, \xi_k\}$
   Compute $\hat{S}_n = \hat{S}_n(\mathbf{g})$, from eq. (6), over the chosen frequency grid
   Compute $Q_n$ the quantile function of $\hat{S}_n$
   Compute $\theta_n^\star$ given by eq. (12) with $Q := Q_n$
---

**Linear computational complexity and quantiles.** The cost of the 2-Wasserstein loss is given by calculating i) the Periodogram $\hat{S}_{\mathrm{Per}}$ of the estimator $\hat{S}_n$ in equation 6, ii) its corresponding quantile function, and iii) the integrals in equation 12. Though the quantile function is available in closed form for some families of PSDs (see Remark 5) they can also be calculated by quantisation in the general case: for a sample distribution (or histogram) one can compute the cumulative distribution and then invert it to obtain the quantile. Additionally, the integrals of the quantile functions also have a linear cost but only in the number of frequency bins $\mathcal{O}(k)$ since they are frequency histograms, therefore, computing the solution has a linear cost in the data.

### 4.3 Solving the general case: beyond the location-scale family

Recall that the proposed method has a closed-form solution when one considers PSDs in location-scale family and the $W_2$ distance over the spectral domain. However, in the general case in equations 7 and 8, the minimisation is not convex in $\theta$ and thus iterative/numerical optimisation is needed. In particular, for horizontal Fourier distances (e.g., Wasserstein distances) the derivative of the loss depends on the derivative of the quantile function $Q_\theta$ with respect to $\theta$, which might not even be known in closed form in general, specially for mixtures. However, in most cases the lags of the empirical covariance or the frequencies of the Periodogram belong to a compact space and thus numerical computations are precise and inexpensive. Therefore, approximate derivatives can be considered (e.g., in conjunction with BFGS) though in our experiments the derivative-free Powell method also provide satisfactory results. Algorithms 2 and 3 present the application of GVM using the temporal and spectral metrics respectively. Note that we will usually consider $L_1, L_2$ as temporal metrics $D$ and also $W_1, W_2$, Itakura- Saito and KL as spectral divergences $D_F$. Algorithms 2 and 3 are presented side to side for the convenience of the reader.

---
**Algorithm 2** GVM - Temporal loss
---
**Require:** $t_i, y_i, i = 1, \ldots, n$
**Require:** parametric family $\{K_\theta\}_{\theta \in \Theta}$
**Require:** temporal metric $D(\cdot, \cdot)$
  Define temporal grid $\mathbf{t} = \{t_1, t_2, \ldots, t_n\}$
  Compute $K_\theta = K_\theta(\mathbf{t})$
  Compute $\hat{K}_n = \hat{K}_n(\mathbf{t})$ as in eq. (5)
  Construct loss $\theta \mapsto D(\hat{K}_n, K_\theta)$
  Find $\theta_n^\star$ in eq. (7) using BFGS or Powell
---

---
**Algorithm 3** GVM - Spectral loss
---
**Require:** $t_i, y_i, i = 1, \ldots, n$
**Require:** parametric family $\{S_\theta\}_{\theta \in \Theta}$
**Require:** spectral metric $D_F(\cdot, \cdot)$
  Define frequency grid $\mathbf{g} = \{\xi_0, \xi_1, \ldots, \xi_k\}$
  Compute $S_\theta = S_\theta(\mathbf{g})$
  Compute $\hat{S}_n = \hat{S}_n(\mathbf{g})$, from eq. (6)
  Construct loss $\theta \mapsto D_F(S_\theta, \hat{S}_n)$
  Find $\theta_n^\star$ in eq. (8) using BFGS or Powell
---

**Linear computational complexity.** For the general case of spectral losses using numerical optimisation methods, we need to calculate $\hat{S}_n$ or its quantile (which are $\mathcal{O}(n)$) only once, to then compute the chosen distance $D_F$, which is $\mathcal{O}(k)$ for discrete measures defined on a $k$-point frequency grid as many times as the optimiser requires it. Therefore, the cost is $\mathcal{O}(k)$ but with a constant that depends on the complexity of the parametric family $\{S_\theta, \theta \in \Theta\}$ and the optimiser of choice. For spatial losses, the complexity follows the same reasoning for the chosen spatial distance $D$ and parametric family $\{K_\theta, \theta \in \Theta\}$, and thus is also linear in the datapoints.

### 4.4 Noise variance and relationship to maximum likelihood

Following the assumptions of the Fourier transform, the spectral divergences considered apply only to Lebesgue-integrable PSDs, which rules out the relevant case of white-noise-corrupted observations. This is because white noise, defined by a Dirac delta covariance, implies a PSD given by a constant, positive, infinite-support, *spectral floor* that is non-integrable. These cases can be addressed with temporal divergences, which are well suited (theoretically and in practice) to handle noise.

The proposed hyperparameter-search method is intended both as a standalone likelihood-free GP learning technique and also as a initialisation approach to feed initial conditions to a maximum likelihood (ML) routine. In this sense, we identify a relationship between the ML estimator $\hat{\theta}_{ML}$ and the proposed estimator $\theta^* := \arg\min_{\theta \in \Theta} L_2(\hat{S}, S_\theta)$. From equation 4 and Plancherel's theorem, we have $\|S - S_\theta\|_{L_2} = \|K - K_\theta\|_{L_2}$. Then, by definition of the estimators and Lemma 2 in Hoffman & Ma (2020), we obtain the following inequality

$$D_{\mathrm{KL}}(\mathbf{K}||\mathbf{K}_{\hat{\theta}_{ML}}) \leq D_{\mathrm{KL}}(\mathbf{K}||\mathbf{K}_{\theta^*}) \leq \frac{1}{2}\|\mathbf{K}^{-1}\|_2\|\mathbf{K}_{\theta^*}^{-1}\|_2\|\mathbf{K} - \mathbf{K}_{\theta^*}\|_F, \tag{13}$$

where $\|\cdot\|_F$ denotes the matrix Frobenius norm, $D_{\mathrm{KL}}(A||B)$ denotes the KL divergence between zero-mean multivariate normal distributions with covariances $A$ and $B$, and recall that $\mathbf{K}$ is the kernel of the ground truth GP. Denoting the ball centred at 0 with radius $M$ by $B(0, M)$, we present the following remark.

**Remark 6.** *The inequality in equation 13 states that if the proposed estimator $\theta^*$ is such that $\|S - S_{\theta^*}\|_{L_2} \in B(0, M)$, then $D_{\mathrm{KL}}(\mathbf{K}||\mathbf{K}_{\hat{\theta}_{ML}}) \in B(0, \frac{1}{2}\|\mathbf{K}^{-1}\|_2\|\mathbf{K}_{\theta^*}^{-1}\|_2 M)$. Therefore, under the reasonable assumption that the function $\theta \mapsto \mathbf{K}_\theta$ only produces well-conditioned matrices, the factor $\|\mathbf{K}^{-1}\|_2\|\mathbf{K}_{\theta^*}^{-1}\|_2$ is bounded and thus both balls have radius of the same order.*

## 5 Experiments

This section illustrates different aspects of the proposed GVM through the following experiments (E):

E1: shows that GVM, unlike ML, is robust to different initialisation values and subsets of observations,

E2: studies the sensibility of GVM to the calculation of $\hat{S}_n$ using Periodogram, Welch and Bartlett,

E3: validates the linear computational complexity of GVM in comparison to the full and sparse GPs,

E4: compares spectral against temporal implementations of GVM on an audio time series,

E5: exhibits the results of learning a 20-component spectral mixture GP using GVM with different spectral metrics,

E6: assesses the ability of GVM to produce initial conditions which are then passed to an ML routine that learns a spectral mixture GP with 4, 8, 12 and 16 components,

E7: presents a toy example where GVM is used to fit a GP with an isotropic SE kernel to multi-input (5-dimensional) data.

The code for GVM is available at `https://github.com/GAMES-UChile/Generalised-Variogram-Method`, a minimal working example of the code is presented in Appendix D. The benchmarks were implemented on MOGPTK (de Wolff et al., 2021).

### 5.1 E1: Stability with respect to initial conditions (spectral mixture kernel, $L_2$, temporal)

This experiment assessed the stability of GVM with respect to random initial conditions and different realisations. We considered a GP with a 2-component spectral mixture kernel $K(t) = \sum_{i=1}^{2} \sigma_i^2 \exp(-\gamma_i \tau^2) \cos(2\pi\mu_i\tau) + \sigma_{\mathrm{noise}}^2 \delta_\tau$ with hyperparameters $\sigma_1 = 2$, $\gamma_1 = 10^{-4}$, $\mu_1 = 2 \cdot 10^{-2}$, $\sigma_2 = 2$, $\gamma_2 = 10^{-4}$, $\mu_2 = 3 \cdot 10^{-2}$, $\sigma_{\mathrm{noise}} = 1$. We produced 4000-point realisations from the GP and 50 random initial conditions $\{\theta^r\}_{r=1}^{50}$ according to $[\theta^r]_i \sim \mathrm{Uniform}[\frac{1}{2}\theta_i, \frac{3}{2}\theta_i]$, where $\theta_i$ is the $i$-th true hyperparameter.

We considered two settings: i) train from $\{\theta^r\}_{r=1}^{50}$ using ML, and ii) compute GVM from $\{\theta^r\}_{r=1}^{50}$ and then perform ML starting from the value found by GVM. Each procedure was implemented using a single realisation (to test stability wrt $\theta^r$) and different realisations (to test stability wrt the data). Our estimates $\hat{\theta}_i$ were assessed in terms of the NLL and the relative mean absolute error (RMAE) of the parameters $\sum_{i=1}^{8} |\theta_i - \hat{\theta}_i|/|\theta_i|$.

Fig. 1 shows the NLL (left) and RMAE (right) versus computation time, for the cases of fixed (top) and different (bottom) observations; all times start from $t = 1$ and are presented in logarithmic scale. First, in all cases the GVM initialisation (in red) took about half a second and resulted in an NLL/RMAE virtually identical to those achieved by ML initialised by GVM, this means that GVM provides reliable parameters and not just initial conditions for ML. Second, for the fixed observations (top), the GVM was stable wrt $\theta^r$ unlike ML which in some cases diverged. Third, for the multiple observations (bottom) GVM-initialised ML diverged in two (out of 50) runs, which is far fewer than the times that random-initialised ML diverged.

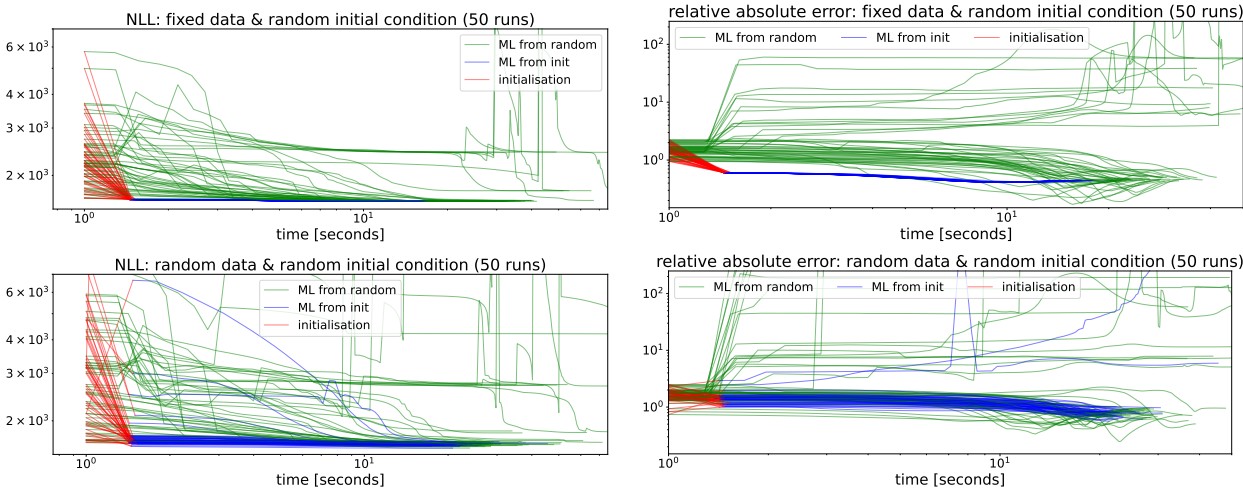

Figure 1: GVM as initialisation for ML starting form 50 random initial conditions: proposed GVM (red), standard ML (green) and ML starting from GVM (blue). The top plots consider a single dataset, while the bottom plots consider different realisations for each initial condition. The L-BFGS-B optimizer was used with the default gradient tolerance in order for the results to be comparable.

## 5.2 E2: Sensibility of GVM wrt the Periodogram (exact case: $W_2$, location-scale)

We then considered kernels with location-scale PSDs and the $W_2$ metric over the PSDs. This case has a unique solution but requires us to compute $\hat{S}_n$ in equation 8; this experiment evaluates different ways of doing so. We produced 4000 observations **evenly-sampled** in $[0, 1000]$ from GPs with square exponential and rectangular PSDs (which correspond to the Exp-cos and Sinc kernels respectively—see Table 1), each with location $\mu \sim U[0.025, 0.075]$ and scale $l \sim U[0.01, 0.02]$. We computed $\hat{S}$ via the Periodogram, Welch and Bartlett methods with different windows. Table 2 shows the percentage relative error (PRE)[3] averaged over 50 runs. The estimates of both parameters are fairly consistent: their magnitude does not change radically for different windows and Periodogram methods. In particular, the estimates of the location parameter are accurate with an average error in the order of 2% for both kernels. The scale parameter was, however, more difficult to estimate, this can be attributed to the spectral energy spread in the Periodogram, which, when using the Wasserstein-2 metric, results in the scale parameter to be overestimated. For both kernels and $\mu = 0.05, l = 0.01$, the case of the (windowless) Periodogram is shown in Fig. 2 and the remaining cases are all shown in Appendix E. In the light of these results, we considered the Periodogram (no window) for the remaining experiments.

---

[3]PRE $= 100 \frac{|\theta - \hat{\theta}|}{\theta}$, where $\theta$ is the true parameter and $\hat{\theta}$ its estimate.

Table 2: Performance of GVM learning GPs with the Exp-cos and Sinc kernels under different sampling settings, Periodogram methods and windows. Each entry of the table shows the average percentage relative error for location (left) and scale (right), with their standard deviations, separated by the symbol "/". True parameters where $\mu \sim U[0.025, 0.075]$ and $l \sim U[0.01, 0.02]$; averages and standard deviations computed over 50 runs.

| Kernel | Window | Periodogram | Bartlett | Welch |
|--------|--------|-------------|----------|-------|
| Exp-cos | none | 2.30±1.61/33.41±13.27 | 2.17±1.91/20.88±11.69 | 2.05±1.64/24.87±14.35 |
| | hann | 2.98±2.14/34.93±14.33 | 3.02±2.36/26.23±13.05 | 2.52±1.75/31.83±13.29 |
| | hamming | 2.92±2.04/34.93±14.16 | 2.93±2.29/27.35±13.26 | 2.49±1.73/32.26±13.22 |
| Sinc | none | 2.36±1.65/8.93±6.73 | 2.63±2.23/83.23±26.39 | 2.31±1.77/38.87±15.95 |
| | hann | 2.68±2.02/10.03±9.22 | 2.59±1.77/58.86±17.48 | 2.44±1.78/19.52±11.32 |
| | hamming | 2.60±2.01/9.62±8.93 | 2.56±1.75/50.62±15.79 | 2.43±1.76/16.51±10.75 |

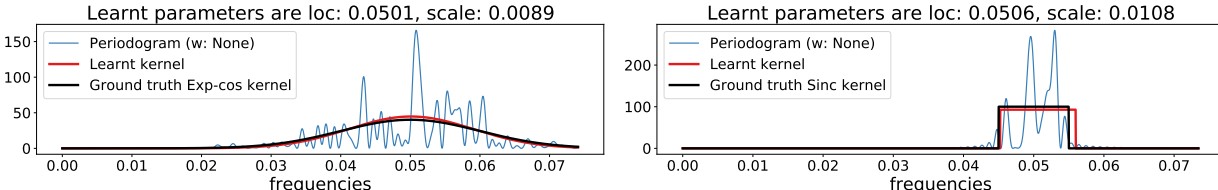

Figure 2: GVM estimates for Exp-cos (left) and Sinc (right) kernels shown in red against $\hat{S}$ (blue) and true kernels (black). This case: Periodogram, no window.

### 5.3 E3: Linear complexity (exact case)

We then evaluated the computation time for the exact case of GVM ($W_2$ distance and location-scale family) for an increasing amount of observations. We considered **unevenly-sampled** observations from an single component SM kernel ($\mu = 0.05, \sigma = 0.01$) in the range $[0, 1000]$. We compared GVM against i) the ML estimate starting from the GVM value (full GP, 100 iterations), and ii) the sparse GP using 200 pseudo inputs (Snelson & Ghahramani, 2006). Fig. 3 shows the computing times versus the number of observations thus validating the claimed linear cost of GVM and its superiority wrt to the rest of the methods. The (solid line) interpolation in the plot is of linear order for GVM, linear for sparse GP since the number of inducing points is fixed, and cubic for the full GP.

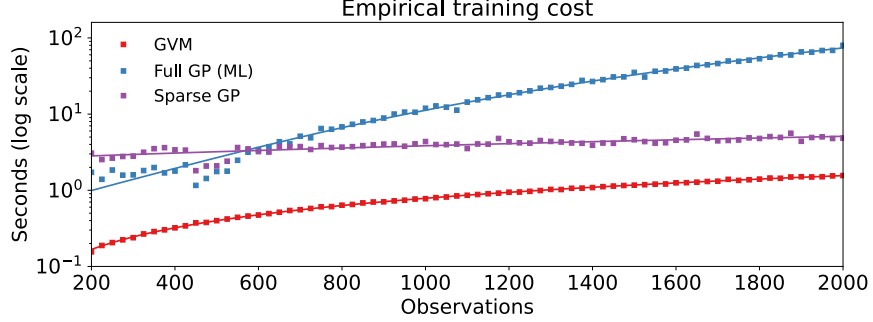

Figure 3: Training times vs number of datapoints for the proposed GVM, full GP and sparse GP.

### 5.4 E4: Performance and cost of spectral and temporal metrics over the same time series

This experiment compares the performance and computational cost of the temporal and spectral implementations of GVM for a common time series and GP models of increasing complexity. We used a real-world

audio signal from the Free Spoken Digit Dataset[4]. GVM was implemented with the metrics $L_1$ and $L_2$ both in the spectral and temporal domain to learn a sample from the above dataset, which was 4300 samples long. The kernel considered was a spectral mixture (SM) (Wilson & Adams, 2013), Fig. 4 shows losses and running times as a function of the number of components of the SM kernel; all runs use the Powell optimiser.

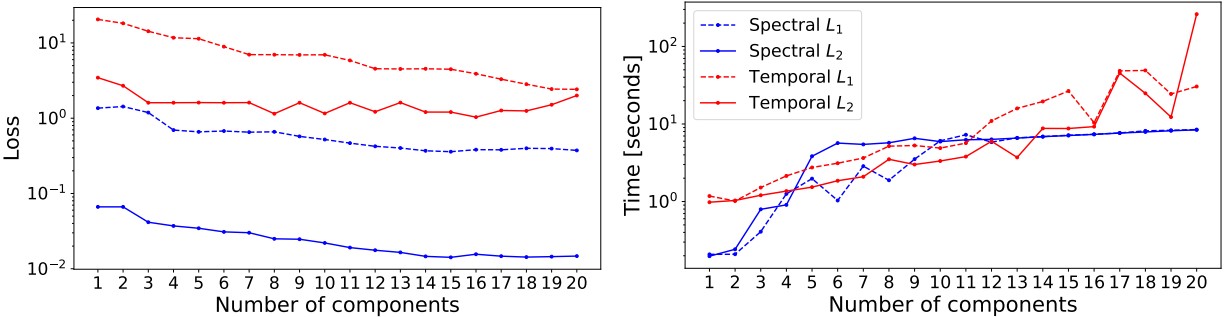

Figure 4: Temporal and spectral implementation of GVM: Losses (left) and running times (right) as a function of the order of the spectral mixture kernel.

From the left plot in Fig. 4 (losses) let us recall that each implementation has its own metric and thus they are not comparable directly, however, notice that both spectral implementations are monotonic with respect to the model order. Furthermore, the fact that the Temporal $L_2$ loss increases with the model order suggests that the optimisation in the time domain is more challenging than in its spectral counterpart. In terms of computational complexity, the right plot in Fig. 4 confirms monotonicity of the computational cost with the number of kernel parameters, however, notice that the spectral implementations reached a plateau after 10 components; further suggesting the superiority of the spectral implementation in terms of ease of optimisation.

Lastly, Fig. 5 shows the fitting of both implementations and metrics in their respective domains. In the time domain (top plots) we can see that the $L_1$ metric (top left) provides a generally acceptable fit but misses some peaks of the autocorrelation function (empirical kernel estimate $\hat{K}_n$), while the $L_2$ metric (top right) aims to reach the peaks of $\hat{K}_n$ at the cost of missing central parts of it. In the spectral domain (bottom plots) we see only minor discrepancies for both metrics, with perhaps the most interesting feature being the difference in the peak at around frequency 0.04, where $L_1$ matched the peak and $L_2$ provided a wider fit.

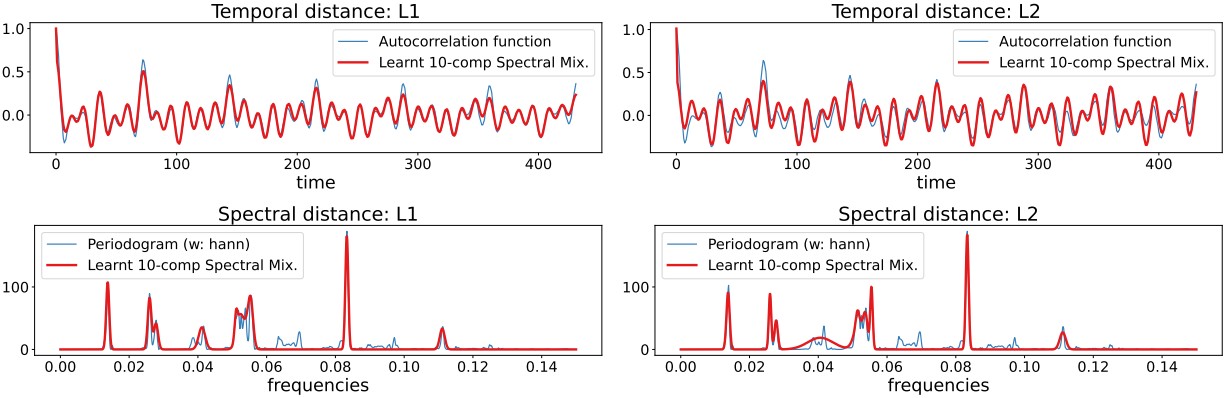

Figure 5: Example of matching for the temporal (top) and spectral (bottom) implementation of GVM for the 10-component SM using the $L_1$ (left) and $L_2$ (right) metrics.

---

[4]https://github.com/Jakobovski/free-spoken-digit-dataset

With this comparison, we validate the intuition that for kernels with concentrated spectral information (such as the spectral mixture) the spectral implementation of GVM is more robust to the metric and faces a less challenging optimisation task.

### 5.5 E5: Fitting a 20-component spectral mixture (different spectral metrics)

This experiment shows the effect of different spectral distances in the GVM estimates, also using an audio signal from the Free Spoken Digit Dataset (different from Experiment 4). Based on the promising performance of the spectral implementation of GVM, we trained a 20-component SM, a kernel known to be difficult to train, and considered the spectral distances $L_1$, $L_2$, $W_1$ and $W_2$ (spectral); Itakura-Saito and KL were unstable and left our of the comparison. Fig. 6 shows the results of the GVM: observe that under almost all metrics, the 20-component spectral mixture matches the Periodogram (considered to be the ground truth PSD in this case). The exception is $W_2$ which struggles to replicate the PSD peaks due to its objective of averaging mass *horizontally*.

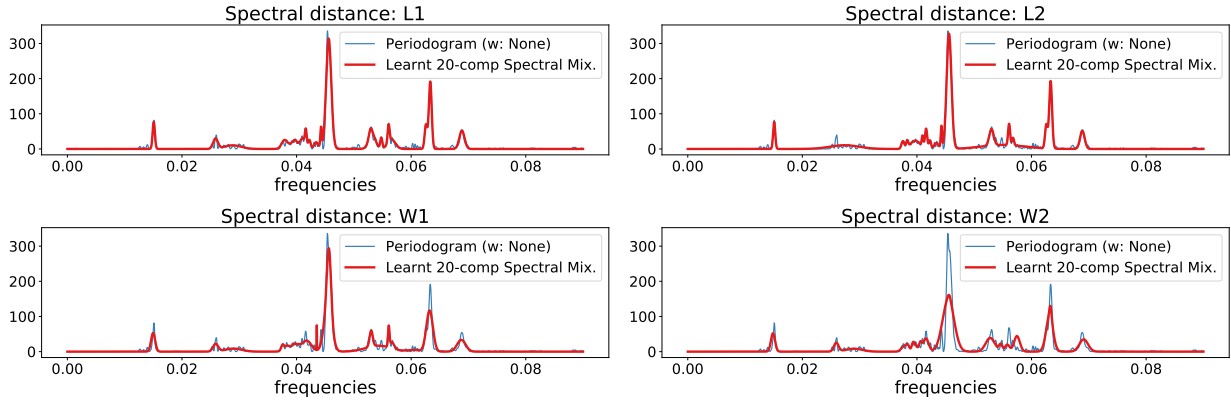

Figure 6: GVM matching Periodogram with a 20-component SM under different spectral metrics.

### 5.6 E6: Learning spectral mixtures and parameter initialisation ($L_2$, multiple orders)

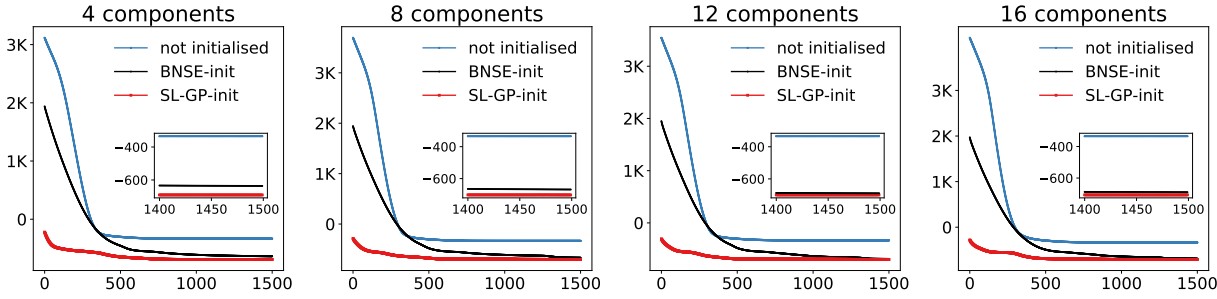

Figure 7: NLLs for the SE spectral mixtures (4,8,12 and 16 components) with different initialisation strategies.

In this experiment, GVM was implemented to find the initial conditions of a GP with SM kernel (4, 8, 12 and 16 components) to a real-world 1800-point heart-rate signal from the MIT-BIH database[5]. We considered the $L_2$ metric (spectral) minimised with Powell and then passed the hyperparameters to an ML routine for 1500 iterations (using Adam with learning rate = 0.1). This methodology was compared against the random initialisation and provided by MOGPTK based on Bayesian nonparametric spectral

---

[5] http://ecg.mit.edu/time-series/

estimation (BNSE) (Tobar, 2018). Fig. 8 first shows the GVM approximations to the heart-rate PSD using 16-component spectral mixtures, for both both kernels.

Table 3: Computation times (secs) for fitting SMs.

|           | 4-comp | 8-comp | 12-comp | 16-comp |
|-----------|--------|--------|---------|---------|
| GVM init  | **2.6**  | **7.6**  | **10.9**  | **23.9**  |
| BNSE init | 74.6   | 68.9   | 72.0    | 74.1    |
| ML        | 420.3  | 464.6  | 529.2   | 582.6   |

Fig. 7 shows NLL for the cases considered. Observe that: i) the non-initialised ML training becomes trapped in local minima in all four cases, ii) the initialisation provided by GVM provides a dramatic reduction of the NLL, even wrt to the BNSE initialisation, iii) the "elbow" at the beginning of the GVM-initialised case suggests that the ML training could have run for a a few iterations (e.g., 100) and still reach a sound solution. Table 3 shows the execution times and reveals the superiority of GVM also in computational time.

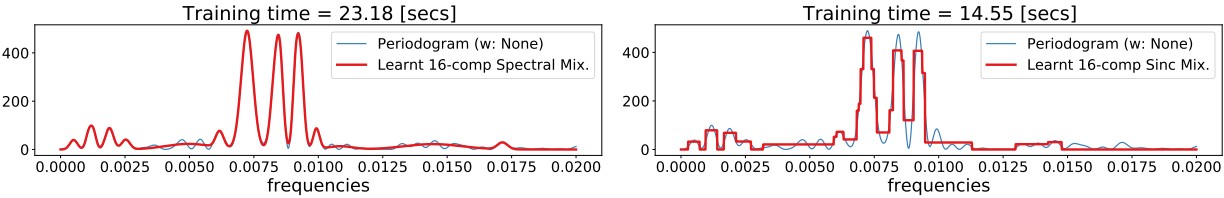

Figure 8: GVM approximations of the PSD of a 1800-sample heart-rate signal using 16-components SE (top) and rectangular (bottom) mixtures. Training time shown above each plot.

### 5.7 E7: Learning an isotropic SE kernel (standard variogram, 5-dimensional inputs, $L_2$)

Though our work focuses on the single-input-dimension case, the rationale behind the proposed GVM method is applicable to datasets of arbitrary input dimension, therefore, to motivate the use of GVM for multi-input GPs, we present a minimal multi-input example. We considered a 5-dimensional GP with SE kernel $K(\tau) = \sigma^2 \exp(-\frac{1}{2l^2}||\tau||^2) + \sigma_{\text{noise}}^2 \delta_t$, and considered four sets of values for the hyperparameters. Notice that we assumed that all 5 dimensions had the same scale parameter, this is the usual setting of the Variogram method in Geostatistics (Cressie, 1993; Chiles & Delfiner, 1999). For each set of hyperparameters, we sampled 1000 points and then implemented GVM with the $L_2$ (temporal) distance to learn the GP. Table 4 shows the estimate error and standard deviation for four sets of hyperparameters averaged over 100 runs, from which the applicability of GVM to address the isotropic multi-input case can be confirmed. For illustration and resemblance to the standard variogram method, Fig. 9 shows the case $\sigma^2 = 5$, $l = 1$, and $\sigma_{\text{noise}} = 1$, where the learnt hyperparameters were $\hat{\sigma}^2 = 5.85$, $\hat{l} = 1.09$, and $\hat{\sigma}_{\text{noise}} = 1.02$.

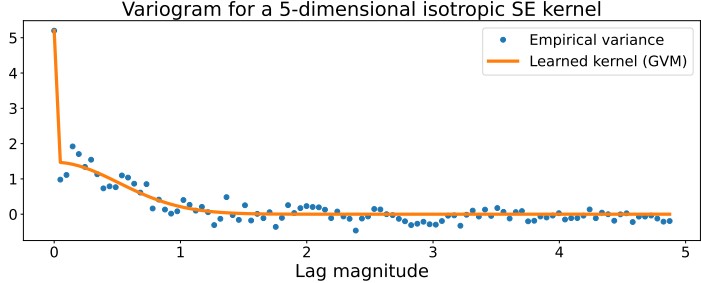

Figure 9: Empirical covariance and fitted covariance function via GVM. The data consisted of 1000 5-dimensional datapoints.

Table 4: GVM applied to multi-input data for four runs of synthetic data using different hyperparameters, i.e., $\sigma^2, l, \sigma^2_{\text{noise}}$. For each set of hyperparameter, $n = 100$ runs were executed to calculate the mean and standard deviation of the estimates (i.e., $\hat{\sigma}^2, \hat{l}, \hat{\sigma}^2_{\text{noise}}$).

| Run | $\sigma^2$ | $l$ | $\sigma^2_{\text{noise}}$ | $\hat{\sigma}^2$ | $\hat{l}$ | $\hat{\sigma}^2_{\text{noise}}$ |
|---|---|---|---|---|---|---|
| 1 | 5 | 2 | 1 | $5.20 \pm 2.07$ | $0.50 \pm 0.09$ | $0.91 \pm 0.68$ |
| 2 | 15 | 2 | 2 | $14.12 \pm 13.14$ | $2.07 \pm 1.09$ | $2.04 \pm 1.51$ |
| 3 | 3 | 3 | 0.5 | $2.68 \pm 3.33$ | $3.24 \pm 1.65$ | $0.97 \pm 0.71$ |
| 4 | 0.5 | 5 | 0.5 | $0.45 \pm 0.62$ | $3.96 \pm 2.50$ | $0.56 \pm 0.30$ |

## 6 Conclusions

By direct minimisation of the discrepancy among covariance functions, we have proposed a novel methodology to pretrain stationary Gaussian processes, which avoids computation of the (cubic cost) likelihood function. The found hyperparameter values can then be passed to an ML-based training routine as initial conditions to conclude the training of the model. Our approach, termed Generalised Variogram Method (GVM), represents a critical improvement in terms of computational complexity: we have shown, both theoretically and empirically, that for the particular case of the 2-Wasserstein spectral distance and location-scale PSDs, GVM is convex and its solution can be computed in a single step. In experimental terms, we have shown the following properties of GVM in the general case: i) applicability to multi-input data, ii) stability wrt to different ways of computing the Periodogram, iii) consistency under different realisations of the GP unlike ML, iv) computational efficiency wrt ML and sparse GPs, v) a realistic alternative to compute initial conditions for ML resulting in considerable reduction of ML iterations, and lastly, vi) ability to train kernels of large number of components that are challenging to train from random initial conditions using ML.

In the general formulation of the proposed GVM (i.e., using either a temporal or spectral divergence) the only requirement in our setup is stationarity, however, particular results in our proposal have specific requirements. First, when using a spectral divergence (the main novel contribution of our work), it is needed that the kernels' Fourier transform (i.e., the PSD) can be computed and is Lebesgue integrable so that $D_F(\hat{S}_n, S_\theta)$ in equation 8 can be calculated; this condition is rather general and admits most stationary kernels used in the literature with the exception of the white noise variance—see Sec.4.4. Second, for the exact case outlined in Thm. 1 we require that the PSD belongs to a location-scale family, this includes standard covariance functions such as the (single-component) spectral mixture, square exponential, sinc, and cosine. PSDs that are not location-scale, such as mixtures of the above kernels, can too be dealt with spectral divergences, however, the solution is not exact and it has to be computed using numerical optimisation methods from equation 8. Third, for all other covariances (including the white noise one) we can use GVM with temporal divergences, which only requires the kernel to be stationary and evaluated pointwise, to find the solution via equation 7.

Spectral kernels are a large class of covariance functions, therefore, the GVM is widely applicable in real-world scenarios in audio, seismology, astronomy, fault diagnosis, finance and any other fields where repetitive temporal patterns arise. In this sense, we hope that our work paves the way for further research in conceptual and applied fields. In theoretical terms, we envision extensions towards non-stationary data using, e.g., time-frequency representations or mini-batches. In practical terms, we aim to address the general-input-dimension case and that our developed companion software will help others make use of GVM as and initialisation method for ML or to directly find the required hyperparameters.

## Acknowledgments

We would like to thank the anonymous referees for their valuable questions and comments, which allowed us to clarify the presentation of the paper and refine our results.

Part of this work was developed when Felipe Tobar was a Visiting Researcher at the *Institut de Recherche en Informatique de Toulouse* (IRIT).

We thank financial support from Google; ANR LabEx CIMI (grant ANR-11-LABX-0040) within the French State Programme "Investissements d'Avenir"; Fondecyt-Regular 1210606; ANID-Basal Center for Mathematical Modeling FB210005; ANID-Basal Advanced Center for Electrical and Electronic Engineering FB0008; and CORFO/ANID International Centers of Excellence Program 10CEII-9157 Inria Chile, Inria Challenge OcéanIA, STICAmSud EMISTRAL, CLIMATAmSud GreenAI and Inria associated team SusAIn.

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

## A  Background: Fourier analysis of continuous-time stochastic processes

We present a brief description of the elements of Fourier analysis used in our proposal. For a more in-depth introduction of the subject, the reader is referred to (Stoica & Moses, 2005) and (Vetterli et al., 2014).

Let us consider a stochastic process over $\mathbb{R}^n$ denoted $(y_t)_{t \in \mathbb{R}^n}$, $n \in \mathbb{N}$. Since the usual condition of stationarity might be restrictive in some cases, we consider the following weaker version of stationarity.

**Definition 3** (Wide-sense stationarity). *The stochastic process $y$ is wide-sense stationary (WSS) if its mean function is constant and its autocorrelation only depends on the temporal difference. That is,*

$$\mathbb{E}[y_t] = \mu(t) = \mu \tag{14}$$
$$\mathbb{E}[y_{t_1} y_{t_2}] = c(t_1, t_2) = c(t_1 - t_2). \tag{15}$$

Though strict stationarity implies WSS, the implication does not hold in the opposite direction. However, for Gaussian processes (GP), whose distribution is fully determined by the first two moments, strict stationarity and WSS are equivalent conditions.

**Definition 4** (Power spectral density). *Let $y$ be a WSS stochastic process with an absolutely integrable correlation function $c(\tau) = \mathbb{E}[y_t y_{t-\tau}]$. The Fourier transform of $c(\cdot)$ given by*

$$S(\xi) = \int_{-\infty}^{\infty} c(\tau) e^{-j 2\pi \xi \tau} d\tau \tag{16}$$

*is called* power spectral density.

Observe that, if both $c$ and $S$ satisfy the conditions for the inversion of the Fourier transform, then $c$ and $S$ are *Fourier pairs*, meaning that we also have

$$c(\tau) = \int_{-\infty}^{\infty} S(\xi) e^{j 2\pi \xi \tau} d\xi. \tag{17}$$

Equations 16 and 17 are a consequence of the Wiener-Khinchin Theorem (Vetterli et al., 2014, p. 292), which relates autocorrelation structure of a WSS process with its distribution of energy across frequencies. This result is instrumental in the construction of GP: under the observation that GPs are uniquely determined by their autocorrelation (or covariance) function, the design of a GP can be conveniently performed in the frequency domain by parametrising the PSD. Furthermore, recall that for zero-mean GPs the autocorrelation and autocovariance functions coincide; thus, we refer to the latter as the covariance kernel.

There are well-known pairs of kernels and PSDs that follow from the above observation. Figure 10 illustrates five cases, showing the covariance kernels, their PSD and a sample of a GP with the corresponding kernel.

In practice, we regard data as realisations of a stochastic process and we need to estimate the covariance or the PSDs from the available datasets. In the signal processing community, these quantities are usually estimated in a nonparametric fashion. For instance, the *sample covariance* in Definition 1 is an estimate of the covariance, which in the case of evenly-spaced dataset $\{y_\Delta, \ldots, y_{N\Delta}\}$ takes the standard form

$$\hat{c}(k\Delta) = \frac{1}{N-k} \sum_{n=1}^{N-k} y_{n\Delta} y_{(n+k)\Delta}. \tag{18}$$

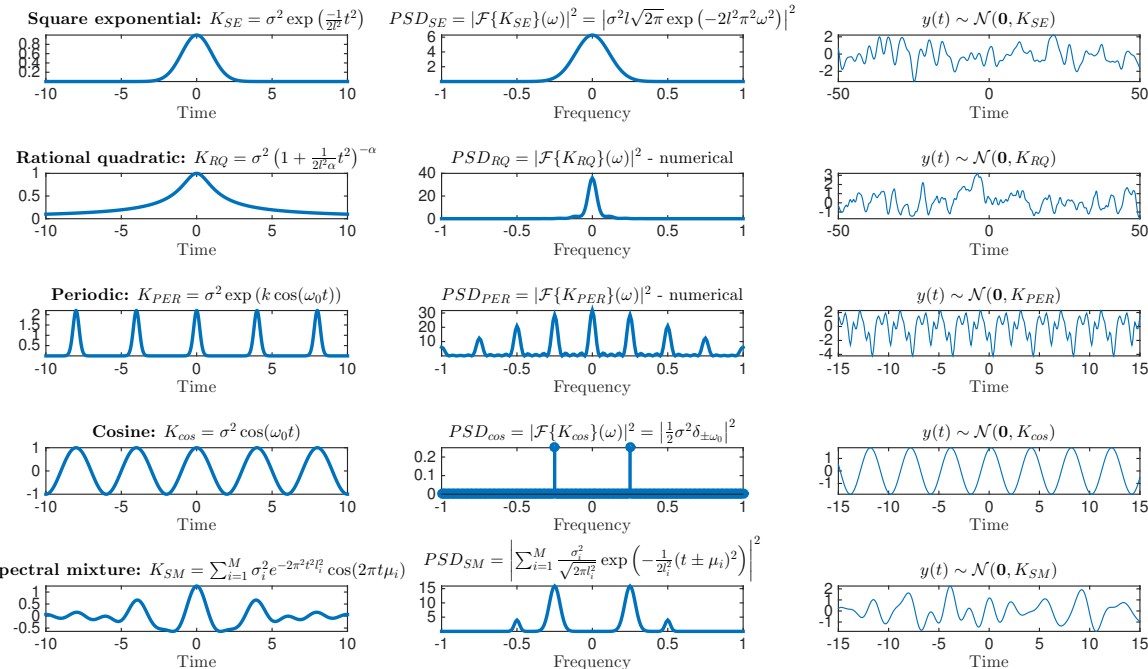

Figure 10: Illustration of the relationship between covariance and spectral representations of GPs. Left to right: kernel, PSD and a GP sample. Top to bottom: square exponential (SE), rational quadratic (RQ), periodic (Per), cosine (cos) and spectral mixture (SM) kernels. For the RQ and Per kernels, the PSD was computed numerically using the discrete time Fourier transform.

The problem of recovering the PSD from a finite collection of observations $\{y_{t_1}, \ldots, y_{t_N}\}$ is referred to as *spectral estimation* and the classic nonparametric method to perform this estimation is called the Periodogram. This technique builds on the observation that, under mild assumptions, the definition of the PSD in equation 16 is equivalent to

$$S(\omega) = \lim_{T \to \infty} \mathbb{E}\left[\frac{1}{2T} \left|\int_{-T}^{T} y_t e^{-j\omega t} dt\right|^2\right]. \tag{19}$$

Therefore, when only a finite dataset of observations is available, the limit and the expectation can be ignored in the above expression thus yielding the natural sample estimate of the PSD given by

$$\hat{S}(\omega) = \frac{1}{N} \left|\sum_{i=1}^{N} y_t e^{-j\omega t}\right|^2, \tag{20}$$

where $N$ in the denominator replaced $2T$ as normaliser.

Equation 20 is known as the *Periodogram*, a widely used method for estimating the PSD of a WSS stochastic process introduced by Schuster (1900). Today, however, the concept of Periodogram refers to a wider class of techniques for spectral estimation that build on the original formulation but at the same time address some of the known drawbacks related to the biasedness and large variance of the Periodogram.

Two widely used extensions of the Periodogram are the Bartlett and the Welch methods. The former operates by splitting the data into segments and computing the standard Periodogram over each one of them, to then average over all computed Periodograms with the aim to reduce the noise in the estimate. The latter follows the same concept but also multiplies each segment by a *window* so as to mitigate the effect of the border of the segments in the estimation. Usual choices for the windows are the Hann and Hamming functions.

## B   Definition of distances and divergences considered

For two functions $f_1$ and $f_2$, we have the following general distances:

- 1-Euclidean : $L_1(f_1, f_2) = \int_{\mathbb{R}} |f_1(\xi) - f_2(\xi)| d\xi$

- 2-Euclidean : $L_2(f_1, f_2) = \int_{\mathbb{R}} (f_1(\xi) - f_2(\xi))^2 d\xi$

Furthermore, when $f_1$ and $f_2$ are densities with quantile functions $Q_1$ and $Q_2$ respectively, we have the additional divergences:

- 1-Wasserstein (Villani, 2009; Peyré & Cuturi, 2019) :

$$W_1(f_1, f_2) = \int_0^1 |Q_1(p) - Q_2(p)| dp$$

- 2-Wasserstein (Villani, 2009; Peyré & Cuturi, 2019) :

$$W_2(f_1, f_2) = \int_0^1 (Q_1(p) - Q_2(p))^2 dp$$

- Kullback-Leibler :
$$D_{\mathrm{KL}}(f_1 \| f_2) = \int_{\mathbb{R}} \log \left( \frac{f_1(\xi)}{f_2(\xi)} \right) f_1(\xi) d\xi$$

- Itakura-Saito (Itakura, 1968) :

$$D_{\mathrm{IS}}(f_1 \| f_2) = \int_{\mathbb{R}} \left( \frac{f_1(\xi)}{f_2(\xi)} - \log \frac{f_1(\xi)}{f_2(\xi)} - 1 \right) d\xi$$

- Bregman divergences (Amari, 2016) : for a function $G : \mathbb{R} \to \mathbb{R}$ that is differentiable and strictly convex,
$$D_G(f_1, f_2) = G(f_1) - G(f_2) - \langle \nabla G(f_2), f_1 - f_2 \rangle$$

Here, we have assumed that both $f_1$ and $f_2$ integrate unity, in the cases where this condition is not met, the densities can be normalised before computing the distance.

## C   Proofs

### C.1   Convexity of spectral loss for $W_2$ and location-scale family

*Proof of Theorem 1.* We recall that

$$W_2^2(S, S_{\mu,\sigma}) = \int_0^1 (Q_{\mu,\sigma}(p) - Q(p)) dp. \tag{21}$$

From a direct application of the rule of differentiation under the integral sign, we obtain the gradient for the location-scale family:

$$\nabla_{\mu,\sigma} W_2^2(S, S_{\mu,\sigma}) = 2 \int_0^1 (Q_{\mu,\sigma}(p) - Q(p)) \nabla_{\mu,\sigma} Q_{\mu,\sigma}(p) dp \tag{22}$$

$$= 2 \int_0^1 (\mu + \sigma Q_{0,1}(p) - Q(p)) \nabla_{\mu,\sigma} (\mu + \sigma Q_{0,1}(p)) dp$$

$$= 2 \int_0^1 (\mu + \sigma Q_{0,1}(p) - Q(p)) \begin{pmatrix} 1 \\ Q_{0,1}(p) \end{pmatrix} dp.$$

The Hessian for the location-scale family:

$$\boldsymbol{H}_{\mu,\sigma}W_2^2(S, S_{\mu,\sigma}) = 2\int_0^1 \begin{pmatrix} 1 & Q_{0,1}(p) \\ Q_{0,1}(p) & Q_{0,1}(p)^2 \end{pmatrix} dp. \tag{23}$$

The determinant of the Hessian (via Jensen's inequality):

$$|\boldsymbol{H}|/2 = \int_0^1 Q_{0,1}(p)^2 dp - \left(\int_0^1 Q_{0,1}(p)dp\right)^2 > \int_0^1 Q_{0,1}(p)^2 dp - \int_0^1 Q_{0,1}(p)^2 dp = 0, \tag{24}$$

where the inequality is strict due to the strict convexity of $(\cdot)^2$.

Therefore, the first order conditions are given by:

$$\int_0^1 (\mu + \sigma Q_{0,1}(p) - Q(p))dp = 0 \iff \mu = \int_0^1 (Q(p) - \sigma Q_{0,1}(p))dp = \int_0^1 Q(p)dp \tag{25}$$

$$\text{and} \quad \int_0^1 (\mu + \sigma Q_{0,1}(p) - Q(p))Q_{0,1}(p)dp = 0 \tag{26}$$

$$\iff \sigma = \frac{\int_0^1 (Q(p) - \mu)Q_{0,1}(p)dp}{\int_0^1 Q_{0,1}(p)^2 dp} = \frac{\int_0^1 Q(p)Q_{0,1}(p)dp}{\int_0^1 Q_{0,1}^2(p)dp}, \tag{27}$$

where in the last expression we have used the fact that the location of the prototype $S_{0,1}$ is zero and so is its mean, meaning that if $x \sim S_{0,1}$ we can write $\int_0^1 \mu Q_{0,1}(p)dp = \mu \mathbb{E}_{x\sim S_{0,1}}[x] = 0$. $\square$

## C.2   Learning from data ($W_2$ distance and location-scale family)

*Proof of Proposition 1.* First, recall that in the location-scale family $\theta = (\mu, \sigma)$. We denote by $Q$ and $\hat{Q}_n$ the respective quantile functions of $S$ and $\hat{S}_n$. Then, following the solutions in Theorem 1 and Jensen's inequality we can compute the following upper bound for the location parameter $\mu^*$:

$$(\mathbb{E}|\mu^* - \mu_n^*|)^2 \leq \mathbb{E}|\mu^* - \mu_n^*|^2 = \mathbb{E}\left|\int_0^1 Q(p)dp - \int_0^1 \hat{Q}_n(p)dp\right|^2$$

$$\leq \mathbb{E}\left[\int_0^1 |Q(p) - \hat{Q}_n(p)|^2 dp\right] = \mathbb{E}[W_2^2(S, \hat{S}_n)],$$

and by hypothesis $\mathbb{E}W_2^2(S, \hat{S}_n) \to 0$.

Similarly, now using Hölder's inequality, we obtain the following bound for the scale parameter $\sigma^*$:

$$\mathbb{E}|\sigma^* - \sigma_n^*| = \mathbb{E}\left|\int_0^1 Q(p)Q_{0,1}(p)dp - \int_0^1 \hat{Q}_n(p)Q_{0,1}(p)dp\right|$$

$$\leq \mathbb{E}\left[\int_0^1 |(Q(p) - \hat{Q}_n(p))Q_{0,1}(p)|dp\right]$$

$$\leq \mathbb{E}\left[\left(\int_0^1 |Q(p) - \hat{Q}_n(p)|^2 dp\right)^{\frac{1}{2}}\right]\left(\int_0^1 |Q_{0,1}(p)|^2 dp\right)^{\frac{1}{2}}$$

$$= \mathbb{E}[W_2(S, \hat{S}_n)]\left(\int_0^1 |Q_{0,1}(p)|^2 dp\right)^{\frac{1}{2}},$$

which tends to 0 by again Jensen's inequality, $(\mathbb{E}[W_2(S, \hat{S}_n)])^2 \leq \mathbb{E}[W_2^2(S, \hat{S}_n)] \to 0$. $\square$

## D  Code

We have developed a short self-contained toolbox which is available in `https://github.com/GAMES-UChile/Generalised-Variogram-Method`. The purpose of the code is to facilitate the use of the proposed method by the community and in particular to replicate all our results. For the reader's convenience, Jupyter Notebook `Exp0_minimal_ex.ipynb` showing a minimal example of the toolbox implementation is presented here.

**Minimal working example of provided code**

```python
[1]: #general imports
     import numpy as np
     import matplotlib.pyplot as plt
     #our package
     from waflgp import *
     import utils
```

```python
[2]: #load data
     signal = np.loadtxt('Data/hr2.txt')
```

```python
[3]: #instantiate model, sum of 16 Gaussians
     q = 16
     gp = waflgp(space_output=signal, aim = 'learning', kernel = 'qSM')#Spectral Mix
     #set frequencies (optional)
     freqs = np.linspace(0,0.02,2000)
     gp.set_freqs(freqs)
```

```python
[4]: #train with periodogram, L2 metric and q components
     gp.train_WL(method = 'periodogram', metric = 'L2', order=q)
     #plot Periodogram and best PSD fit
     gp.plot_psd(title = f'Minimal example')
```

```
Optimization terminated successfully.
         Current function value: 0.002876
         Iterations: 33
         Function evaluations: 21261
L2-ok
```

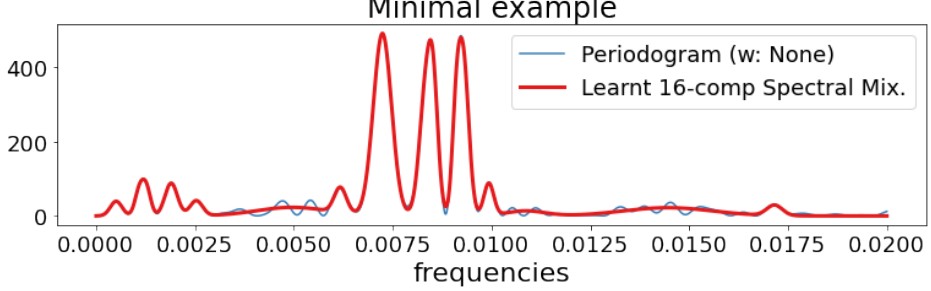

# E   Additional figures for E2

Experiment E2 showed the sensibility of GVM to the choice of Periodogram method and window for two kernels. Figures 11 and 12 presents all the figures corresponding to the estimates in Table 2 for the Exp-cos and Sinc kernel respectively.

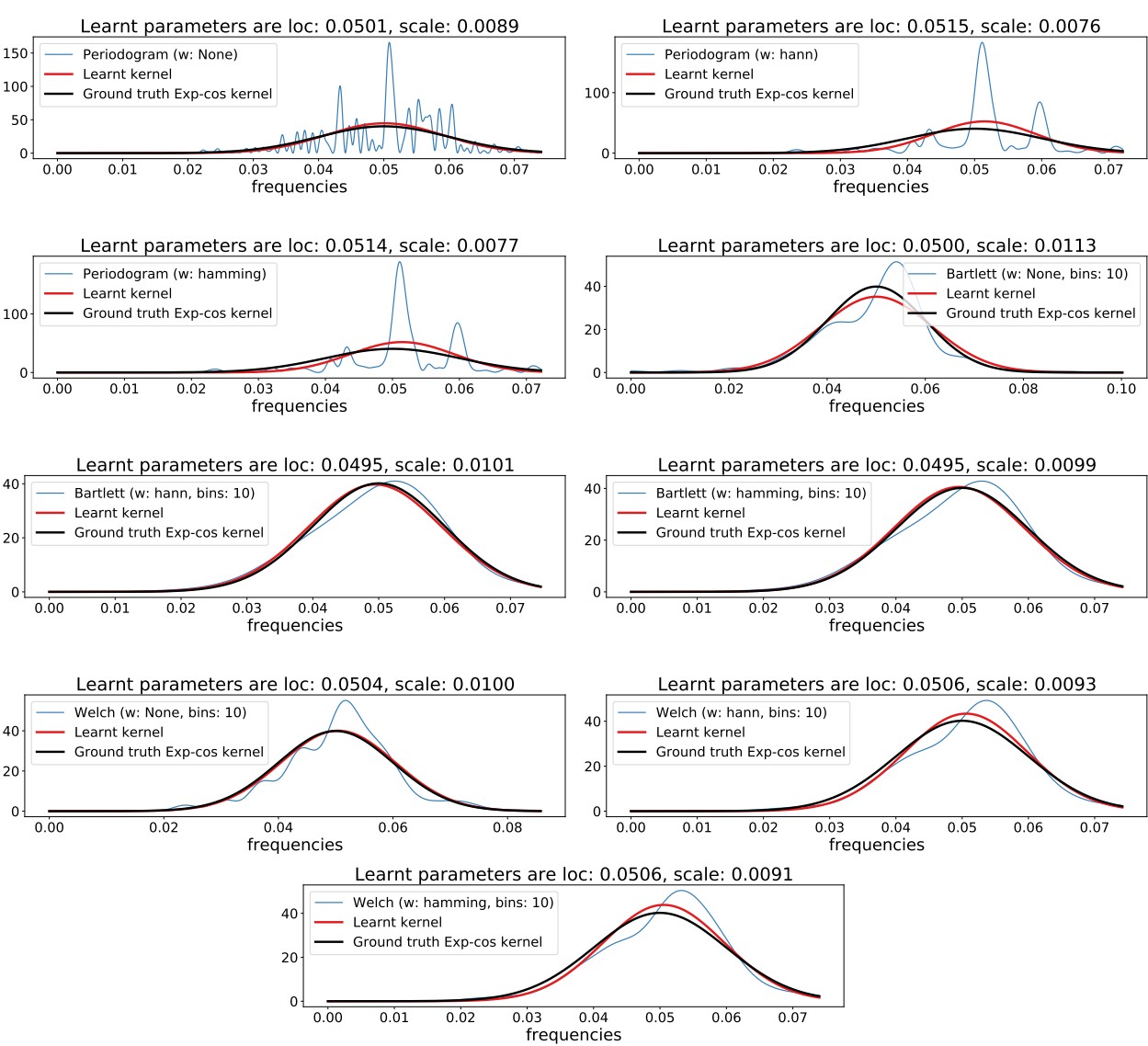

Figure 11: Exp-cos kernel.

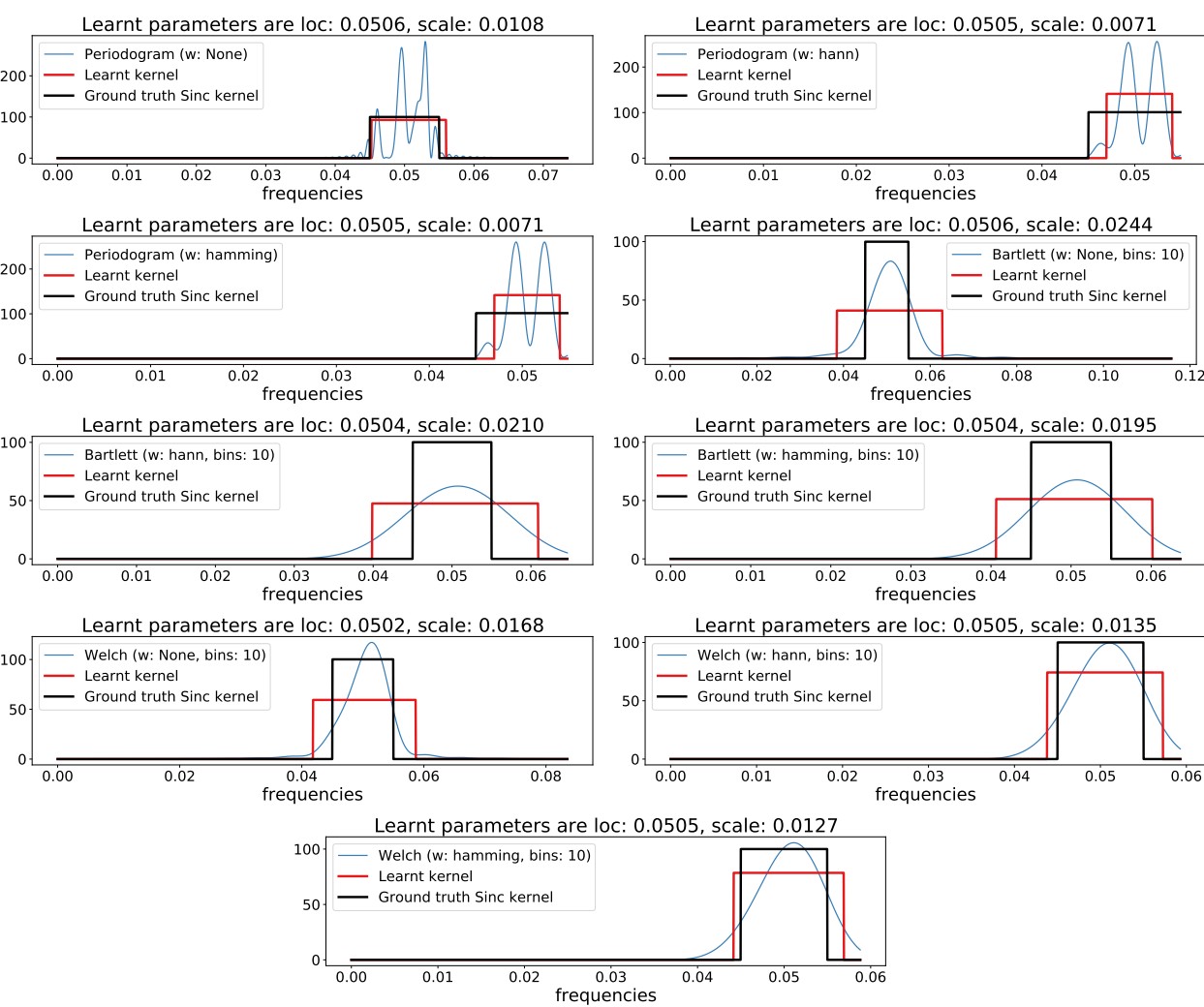

Figure 12: Sinc kernel.

