# OpenReview forum: "Computationally-efficient initialisation of GPs: The generalised variogram method"
_TMLR — Accepted by TMLR_

### Review · Reviewer_VErd · 2022-11-19

**Summary Of Contributions:**

This work starts with the observation that maximizing likelihood for GP is equivalent to minimizing KL-divergence between the true GP and the GP to be learnt. Then it looks for methods to find GP hyperparameters by minimizing a loss function on 1) sample covariance and 2) its Fourier transform through power spectral density (PSD).

The author proved, for location scale PSD and W2 distance, one can find the optimal location and scale parameters analytically (Theorem 1) and it converges to the true minimizer (Proposition 1) and also in more general cases under certain conditions (Proposition 2).

The empirical results show that the proposed method can recover the true hyperparameter with good accuracy (Sec. 5.1), stable with initializations (Sec. 5.2), sensibility (Sec. 5.3), scalability (Sec. 5.4) and kernel mixtures on real-world time series data (Sec. 5.5 and 5.6).


**Audience:**

Yes

**Broader Impact Concerns:**

No.

**Claims And Evidence:**

Yes

**Requested Changes:**

I would suggest the authors:
- Adding results using eq(4) and comparing to the PSD based methods.
- Making the paper more self-contained, see some examples in the weakness section.
- Explicitly discussing the focus and the limitation of the methods. Whether it focuses on time series (1d) data and what kernels can not be supported currently and what is the implication of that.
- Including kernel alignment as variants for eq(4) and even better, providing empirical results.


**Strengths And Weaknesses:**

### Strength:
It’s a novel and interesting idea. The proposed method provides an alternative for learning GP hyperparameters. The main advantage of the method is the scalability (linear time complexity) and stableness to random initialization.

The work has both theoretical and empirical contributions. The experiments are organized with questions, making it easier to consume.

### Weakness:

A **critical** point is that I didn’t see any results using eq(4) (please correct me if I am wrong). To show learning GP hyperparameters in the spectral domain helps, it is critical to compare the proposed method to that baseline.

Clarity and self contained writing.
I find it difficult to build the context when reading the paper, partially due to the fact that I am less familiar with Fourier Transform and not using them much in my own work. But I think the authors could also make the paper more self contained for the readers in a slightly different area. Below I give some examples:

- In Theorem 1 where the PSD is restricted to location scale family and using W2 distance, the optimal location and scale parameter can be found in one step. But how to get back to the original kernel hyperparameter (I assume by using inverse Fourier transform)? Stating it explicitly, even better with some examples, could help.

- In the end of Proposition 2, the author mentioned “windowing and the Welch/Bartlett technique” and they are used in Table 1, but they are never explained in the text.
- In Section 4.1, authors briefly state how to handle general cases but in my opinion, they would be better presented in pseudo code with clear input and output.

- In Section 5.5, how to get the groundtruth Periodogram for the Free Spoken Digit Dataset?


Limitation: The method only supports PSD with a specific parametric family and explicit inverse Fourier transform. This suggests, as stated also by the authors in the end of Section 3.1, a set of 6 kernels (Dirac delta, Cosine, Square Exponential (SE), Student’s t, Sinc, Rectangle) and their mixtures can be covered. Is this a complete list? The author should discuss this and its impact on real-world applications explicitly.

Time series data. The experiments are mostly 1-d input time series data. The only exception is in Section 5.1, but it is a synthetic one. It would be nice to have more real-world multi-dimensional input regression tasks. If the author focuses on time series data, that should be stated explicitly in the contribution.

 Also, there is a whole area of kernel alignment and it can be seens as a variant of eq(4). A good pointer for kernel alignment is:

Cortes, Corinna, Mehryar Mohri, and Afshin Rostamizadeh. "Algorithms for learning kernels based on centered alignment." The Journal of Machine Learning Research 13 (2012): 795-828.

---

> ### Comment · Reviewer_VErd · 2023-02-10
> **Updated review after 1st paper revision**
>
> I would like to thank the authors for their revisions and kindly marking the changed part with red color for the ease of reading. There are two major concerns from my side:
>
> - Is there some results from Algorithm 1? As mentioned in the paper, one could optimize kernel hyperparameters in both temporal domain and frequency domain for eq(5). But there is no comparison to the methods in the temporal domain (for eq(5)). I can imagine finding kernel hyperparameters in the frequency domain could be more efficient and robust, but that needs to be shown. Also, the kernel alignment becomes relevant for methods in the temporal domain and I would hope to see some results there as well.
>
> - Clarity. I am aware the code is given, but as a test for clear writing, I still don’t think I could implement the algorithm based on the given  description. Many details are missing. For example (not limited to it), in the last line of Algorithm 1 & 2, “either numerical methods or gradient methods are used”. What does the author mean by numerical methods? Is it to use finite differences as an approximation of the gradient?  is it the derivative-free Powell method? As another example, when it comes to exact method, what is the quantile function Q_{0, 1}(q) for some common kernels? Could you give the math on how to compute the the solution for some common kernel?

---

### Review · Reviewer_dRzy · 2022-11-25

**Summary Of Contributions:**

The authors consider how to initialize the hyperparameters of (time-series) GPs using ideas related to but extending the variogram. Both theoretical results and practical implementations are presented, first for a more restricted setting (location-scale power spectral densities, PSD [an unfortunate acronym!] with 2-Wassertein distance) and then for more general setups. A nice series of experiments of increasing complexity are conducted, from a simple squared exponential kernel initialized via the variogram to complex spectral mixtures.


**Audience:**

Yes

**Claims And Evidence:**

Yes

**Requested Changes:**

I would have liked some more clarity around what results apply to multi-input settings (and how). The intro and math setup is very focused on time-series analysis, but the experiments section then starts out with E1 being a multi(=5) input example! My understanding is that for E1 the lengthscale is shared across the input dimensions: can this be relaxed to one per dimension? In that setting smart initialization would seem particularly important because there are potentially many hyperparameters.

For E1 and E3 the experiments should vary the true hypers (at least l and mu, maybe sigma) over some range – otherwise for all we know GVM always returns a number around 1 in the E1 setting which just happens to be the "correct" answer. Then the plot of true vs estimated hyperparameter could be inspected.

I've annotated minor comments on the pdf here:
https://www.dropbox.com/s/shoogfeux3th8gs/computationally_efficient_init.pdf?dl=0

**Strengths And Weaknesses:**

The presentation is reasonably clear although it does presume quite a lot of material (e.g. several kernels that are not defined such as sinc). Given the page limit this seems difficult to address. The bigger challenge I had with the exposition was keeping track of what theoretical and experimental results apply to what approaches. I suspect there is some nice summary that could be provided: maybe a table where the rows are different approaches and the columns are different properties (e.g. convexity)?

Other than the minor concerns below I'm positive about the paper: good initialization is one of the important but overlooked aspects of statistical ML, and the theory and empirical results here should both be useful to the field.

---

### Review · Reviewer_Z3Qg · 2023-01-31

**Summary Of Contributions:**

Inspired in signal processing/geostatistics methods, the paper proposed a new technique for learning the kernel hyperparameters in Gaussian process (GP) models. The key idea behind is to avoid the evaluation/computation of the likelihood function. To do that, the paper first makes an observation on the similarity between the (expected) log-marginal likelihood  of the GP and the negative KL divergence between two multivariate Gaussian distributions. In particular, this leads to consider distance/divergence based methods between the kernel-derived covariance matrix and the true empirical one. The focus is then placed into both distance based i.e. L1, L2 norms and spectral metrics (between the PSD of the true and fitted covariance matrices). In practice, estimating both leads to methods that are promising when learning the hyperparameters, both in computational cost and performance (reaching the true ones).


**Audience:**

Yes

**Broader Impact Concerns:**

I do not detect particular ethical implications of the work.

**Claims And Evidence:**

Yes

**Requested Changes:**

To me the paper has three main points that should be fixed or at least improved:

- Establish new connections between different approaches is generally difficult, and usually makes one to incur in odd mix of notations, concepts, ideas, etc. In this case, the task would be between the general approach of learning GP hyperparameters, spectral methods in signal processing and divergences/distances. In my opinion, this could be improved significantly, mainly in Sections 3 and 4, where the goal of learning hyperparameters is somehow secondary behind the presentation of several techniques and ideas.

- Related work and a deep review of the papers working on the learning of GP's hyperparameters -- this at the end, should be also accompanied by extra experiments, proving the similarities and dissimilarities between the proposed method and previous efforts in SOTA. On this, I want also to remark that the results seem promising to be, but not entirely well presented in the GP literature framework.

- I would like to see the presentation of the computational cost, algorithmic solution better presented or at least clearer for the authors. If it is computationally cheap and fast, one should feel that is easy to code or implement.

**Strengths And Weaknesses:**

** Strengths **

In my opinion, the paper tackles an interesting idea for a well-known problem in the GP community. There has been always a general interest and intuition around the idea that spectral methods or signal-processing inspired tools could solve many of the issues that GP learning usually faces. In this case, I enjoyed the spirit behind the connection between the expected log-marginal likelihood and the divergences, since it opens new doors to the problem of learning hyperparameters. The general idea of bringing PSD/Kernel distances to be used as a target loss to minimise in order to reach meaning results makes sense to me as well. Some of the experimental results on toy data also seem promising.

** Weaknesses **

Even while I enjoyed reading the paper, there are points or ideas that are not entirely explained or extremely clarified. Some other details or experimental results seem somehow limited to me, in the sense that they do not compare with other SOTA methods who have also tried to solve this problem (see references below).

- One example, for instance is the connection between the KL divergence and the expected log-marginal likelihood. The claim that both quantities are equivalent is not entirely true to me. First of all, there are terms weighted in a different manner, and the quantities are not the same at all. Also, the expected log-marginal likelihood is not the quantity of interest for us from a Bayesian perspective, right? Since we are observing a single training dataset. Later on, the use of divergences between covariance matrices seems odd to me, or at least I would expect this sort of metrics being computed wrt probability densities, not covariances instead.
- Another weak point is Section 3 and 4, where there is a bit of lack of clarity. From the reader's perspective, one feels that the thread of the paper fuses too many concepts from different sides, but not really contributing to the central idea of the paper that is the learning of hyperparameters. For instance, the presentation of Algorithm 1 and Algorithm 2 is somehow very light, and not very reproducible from the perspective of a computer scientist. How are exactly this losses computed, how is the iteration through the data on this algorithm? We would need answers like that.
- On the experimental side, I find that the results are promising, but somehow preliminary to me. As a first view/intuition of the performance of the method is right, but to really be a mind-changing method for the GP community, one should include rigorous comparisons with other methods in the literature. Also playing with larger data, different datasets (i.e. UCI datasets at least) and other well-known benchmarks... that would be the most convenient.
- Finally, from the contribution aspect, one might easily feel that the proposed solution are L1,L2, and Wasserstein distances between estimated covariances and PSD functions, which itself is fine, but I feel that some of them could have been partially considered in the very long literature of GPs since its boom in the ML community after 2006. At least, a larger review would be required to be entirely trusted, in my opinion.


** References **

1] V. Lalchand et al. "Sparse Gaussian Process Hyperparameters: Optimize or Integrate?" NeurIPS 2022

2] S. Liu et al. "Task-Agnostic Amortized Inference of Gaussian Process Hyperparameters" NeurIPS 2020 (see related work in this paper and references to approaches for learning GP hyperparameters).

3] D. Burt et al. "Rates of convergence for sparse variational gaussian process regression" ICML 2019 (I think this one is related to what is proposed in the paper, even if it focuses on sparse methods. I say this bc authors here use the Nystrom approximation to the data covariance matrix and a bound on the KL divergence between the true posterior and the approximated one.)

4] Rassmussen et al. "Gaussian Processes for Machine Learning" 2006 - Ch. 5 including the references to cross validation and other related methods.

---

### Decision · Action_Editors · 2023-04-17

**Recommendation:** Accept as is

**Comment:**

All reviewers agreed on the technical novelty of the paper. The reviewers were also happy with the updates made by the authors and the additional supportive evidence they provided. This work is of interest to the community so I am supporting acceptance as is -- this is the third revision provided by the authors, so the paper is pretty polished.

**Audience:**

This paper is of interest to the TMLR audience. In particular, the paper makes contributions to Gaussian process and Bayesian machine learning subcommunities.

**Claims And Evidence:**

All reviewers recommended acceptance after the revisions made by the the authors. The claims are supported, concerns were addressed and the manuscript was significantly improved following the suggestions of the reviewers.